# Transcriptomic and Proteomic Analysis of *Gardnerella vaginalis* Responding to Acidic pH and Hydrogen Peroxide Stress

**DOI:** 10.3390/microorganisms11030695

**Published:** 2023-03-08

**Authors:** Kundi Zhang, Mengyao Lu, Yuxin Qiu, Xiaoxuan Zhu, Hongwei Wang, Yan Huang, Hongjie Dong, Lichuan Gu

**Affiliations:** 1State Key Laboratory of Microbial Technology, Shandong University, Qingdao 266237, China; 2Shandong Institute of Parasitic Diseases, Shandong First Medical University & Shandong Academy of Medical Sciences, Jining 272033, China

**Keywords:** *Gardnerella vaginalis*, lactobacilli, lactate and hydrogen peroxide, transcriptomics, biofilm formation

## Abstract

*Gardnerella vaginalis* is the main pathogen that causes bacterial vaginosis. In the healthy vaginal microecological environment of a woman, the lactobacilli produce lactate and hydrogen peroxide to inhibit the growth of pathogens such as *G. vaginalis*. The lack of lactobacilli results in a high pH and low hydrogen peroxide in the vagina which facilitate *G. vaginalis* growth, leading to the imbalance of the vaginal microecology. In this study, lactate and hydrogen peroxide were added to a *G. vaginalis* culture medium to simulate the co-culture of the lactobacilli and *G. vaginalis*, and then the genes related to the stress response of *G. vaginalis* were identified using transcriptomics and proteomics. It was indicated that, among all the upregulated genes, most of them encoded transporters associated with the efflux of harmful substances, and the majority of the downregulated genes were related to the biofilm formation and epithelial cell adhesion. This study may help find new drug targets for *G. vaginalis* for the development of novel therapies for bacterial vaginosis.

## 1. Introduction

The reproductive tract is an open cavity that houses a large number of microorganisms and is closely related to female reproductive health. It is a dynamic environment in which drug stimulation, host hormone levels and immunity affect the microbiome composition, resulting in a microecological imbalance and bacterial vaginosis (BV) [1].

A loss of vaginal lactobacilli and an overgrowth of diverse anaerobes are two characteristics of a vaginal microecological imbalance. To cause BV, the anaerobic microbiota produce enzymes (such as sialidase) that degrade the mucosa, produce biofilm and secrete short-chain fatty acids [2]. BV is associated with pelvic inflammatory disease [3] and infection following gynecological surgery and cervical cancer. Patients can develop complications whether or not they have clinical symptoms. BV in pregnancy increases the risks of spontaneous abortion, amniotic chorionitis, premature rupture of the membranes and premature delivery, all of which have serious consequences for a woman’s health [4,5]. The incidence rate of BV is approximately 23–29%, with significant differences between ethnic groups [6,7]. The standard treatment for BV is metronidazole and clindamycin, and the recurrence rate is as high as 30% within 3 months [8]. Probiotics therapy has recently achieved some success. The vaginal flora of healthy women is dominated by the lactobacilli while patients with BV have a diverse microbiota containing anaerobic and uncultivable species [9]. In clinical and laboratory features, vaginal microbiome transplantation from healthy donors resulted in four patients obtaining full long-term remission and one patient achieving incomplete remission [10]. A study of *Lactobacillus crispatus* CTV-05 after vaginal metronidazole treatment found a significantly lower incidence of BV recurrence [11]. Even *Saccharomyces cerevisiae*-based probiotics obtained beneficial effects for preventing and/or treating vulvovaginal candidiasis (VVC) and BV [12]. However, effective prevention and treatment strategies must still be developed.

*Gardnerella vaginalis* plays an important role in the establishment of the BV bacterial community, for which it can adhere to the host epithelium, produce biofilm and secrete vaginolysin [13]. In a healthy vagina, the lactobacilli directly inhibit the growth of *G. vaginalis* by secreting lactate, hydrogen peroxide and bacteriocin, and promote the integrity of the epithelial cells by stimulating the mucus secretion and regulating the immune response [14].

To date, how lactate and hydrogen peroxide stress affect *G. vaginalis* and how *G. vaginalis* responds to stress are unknown. To simulate the co-culture of the lactobacilli and *G. vaginalis*, we cultivated *G. vaginalis* in the presence of lactate and hydrogen peroxide. As a deprived species in cell signal transduction, there were no c-di-A/GMP and only six two-component system (TCS) genes in the genome of *G. vaginalis* [15]. Therefore, the *shk* (sensor histidine kinase), which encodes a membrane protein belonging to the TCS and is related to the regulation of the biofilm formation [16], and the *trx* (thioredoxin disulfide reductase), which encodes an intracellular protein responding to lactic acid and hydrogen peroxide [17], were selected as the markers to indicate the exact moment when the cells exhibited the strongest response. Transcriptomics and proteomics were then used to identify the genes involved in the *G. vaginalis* response to the acidic pH and hydrogen peroxide stress.

## 2. Materials and Methods

### 2.1. Strains and Growth Conditions

*G. vaginalis* ATCC 14,019 was purchased from the American Type Culture Collection. The planktonic cells were grown in an sBHI [10.0 g of peptone, 12.5 g of dehydrated calf brain extract, 5.0 g of dehydrated beef heart extract, 5.0 g of NaCl, 2.0 g of glucose, 2.5 g of disodium hydrogen phosphate in 1 L ddH_2_O, pH 7.4] for 12 h at 37 °C with 5% CO_2_.

### 2.2. Growth Curve Test

A two-percent seed solution was inoculated into the sBHI without sheep’s blood and cultured to the early exponential phase when the cell density reached an OD_600_ of approximately 1.0. At this point, the lactate (Sigma Aldrich L6661, Shanghai, China) and hydrogen peroxide (Sigma Aldrich H1009, Shanghai, China) were added to the pH 5.5 and 0.5 mM, respectively, to obtain the sub-lethal cells [18,19]. Then, 5 min, 10 min, 30 min and 1 h after the start of stress, the culture samples were collected to measure the cell density and transcription levels of the response marker genes, including the *shk* and *trx*. The transcriptome and proteome analyses were performed on the samples with the highest level of the response marker gene transcription. Three biologic replicates were carried out for each condition.

### 2.3. cDNA Library Preparation and Sequencing

A total amount of 3 μg of RNA per sample was prepared as the input material for the RNA sample preparations. A Vazyme Ribo-off rRNA depletion kit (bacteria) (Vazyme biotech, Piscataway, NJ, USA) was used to remove rRNA. The sequencing libraries were generated using an NEB Next UltraTM RNA library Prep Kit (NEB, Ipswich, MA, USA).

After purification, the RNA samples were fragmented into small pieces using divalent cations under an elevated temperature. The cleaved RNA fragments were reversely transcribed into first strand cDNA using random primers. The strand specificity was achieved by replacing dTTP with dUTP in the Second Strand Marking Mix (SMM), followed by a second strand cDNA synthesis using DNA Polymerase I and RNase H. The incorporation of dUTP in the second strand synthesis quenched the second strand during the amplification since the DNA polymerase used in the assay did incorporate nucleotide after dUMP. The addition of Actinomycin D to the First Stand Synthesis Act D mix (FSA) prevented spurious DNA-dependent synthesis, while allowing for RNA-dependent synthesis and improving the strand specificity. These cDNA fragments were then added to a single ‘A’ base at the 3′ end for subsequent ligation to the adapter. The products were then purified and enriched with PCR to create the final cDNA library. The library quality was assessed using the Agilent Bioanalyzer 2100 system. The library preparations were sequenced on an Illumina Hiseq 4000 platform by the Beijing Allwegene Technology Company Limited (Beijing, China) and the paired-end 150 bp reads were generated.

### 2.4. RNA-Sequencing Data Analysis

The raw data (raw reads) of the fastq format were firstly processed using in-house perl scripts. In this step, the clean data (clean reads) were obtained by removing the reads containing adapter, the reads containing ploy-N (N > 10%) and the low-quality reads (Q < 5) greater than 50% from the raw data. At the same time, the Q20, Q30 and GC content of the clean data were calculated. All the downstream analyses were based on the clean data with a high quality. The adaptor sequences and low-quality sequence reads were removed from the data sets. The raw sequences were transformed into clean reads after data processing. These clean reads were then mapped to the reference genome sequence by Bowtie2 v2.2.6. Only the reads with a perfect match or one mismatch were further analyzed and annotated based on the reference genome. The Bowtie2 mapping results were assembled using the Rockhopper software v2.0.3 and compared to the annotated gene models to find new transcript regions. The Blastx program was compared to the NR library (the e-value is set to le-5). The newly predicted transcript region was annotated and the annotated transcript region was regarded as a new transcript region with coding potential.

### 2.5. Quantitative PCR

The total RNA was extracted using the MiniBEST kit (TaKaRa, Dalian, China) and was reverse transcribed using the TransScript II One-Step gDNA Removal and cDNA Synthesis SuperMix kit (Transgen Biotech, Beijing, China). In order to confirm the stress response time of the strain and validate the RNA-seq data, a qPCR was performed to quantify the transcription level of the signal transduction genes and virulence genes, respectively. The *gyr* gene was used as the reference gene. The oligonucleotide primers were designed using SnapGene (Appendix A). The qPCR reaction was prepared by mixing together 10 μL of SYBR green supermix (Bio-Rad, Hercules, CA, USA), 1 μL of 1:800 diluted cDNA, 0.5 μL of 10 μM forward and reverse primers and water up to 20 μL. The run was performed using a CFX96TM thermal cycler (Jena, Germany) with the following cycling parameters: 3 min at 95 °C, followed by 40 cycles of 10 s at 95 °C, 10 s at 60 °C and 15 s at 72 °C. The normalized gene expression was determined using the delta Ct method (EΔCt), a variation of the Livak method, where ΔCt = Ct (reference gene)−Ct (target gene) and E stands for the experimentally determined reaction efficiency. Three biologic replicates of each condition were analyzed.

### 2.6. Relative Analysis of the TMT-Marked Proteins

For each sample, a 100 μg peptide mixture was labeled using the TMT reagent according to the manufacturer’s instructions (Thermo Fisher Scientific, Waltham, MA, USA). Each fraction was injected for a nano LC-MS/MS analysis. The peptide mixture was loaded onto a reverse phase trap column (Thermo Scientific Acclaim PepMap100, 100 μm * 2 cm, nanoViper C18) connected to the C18-reversed phase analytical column (Thermo Scientific Easy Column, 10 cm long, 75 μm inner diameter, 3 μm resin) in buffer A (0.1% formic acid) and separated with a linear gradient of buffer B (84% acetonitrile and 0.1% formic acid) at a flow rate of 300 nL/min controlled by the IntelliFlow technology. The LC-MS/MS analysis was performed on a Q Exactive mass spectrometer (Thermo Fisher Scientific) that was coupled with the Easy nLC (Proxeon Biosystems, now Thermo Fisher Scientific) for 60/90 min (determined by the project proposal). The MS/MS spectra were searched using the MASCOT engine (Matrix Science, London, UK; version 2.2) embedded into Proteome Discoverer 1.4.

## 3. Results

### 3.1. Acidic pH and Hydrogen Peroxide Significantly Inhibit G. vaginalis Growth

To determine the effect of the acidic pH and hydrogen peroxide stress on *G. vaginalis* development, the cell growth curves for *G. vaginalis* were obtained in the sBHI medium, Sbhi + acidic pH and sBHI + acidic pH + hydrogen peroxide. The three cultures had nearly identical growth curves during the lag and initial log phases. After adding lactate to one group of samples at 5 h to adjust the pH to 5.5, the cells grew slower for about 2 h, then declined for 3 h before beginning to recover. In the transcriptomic and proteomic analysis, these samples were named as the Lac group. When both lactate and H_2_O_2_ were added to another group of samples to achieve a pH of 5.5 and a final concentration of 0.5 mM of H_2_O_2_, the cells grew identically to the Lac group for 1 h before entering the stable phase with a cell density lower than the stress-free control group. In the following study, the samples subjected to the acidic and oxidative stress were labeled as the LacHyd group (Figure 1A). The expression levels of the response marker genes *shk* and *trx* peaked 10 min after the stress began, according to the RT-PCR data. After one generation time (1 h), the expression level of these genes began to decline and reached its lowest resting level (Appendix A). Therefore, the Lac and LacHyd group 10-min cultures were subjected to the transcriptomics and proteomics analysis.

### 3.2. Genes Related to Biofilm Formation and Epithelial Adhesion Were Downregulated during Stress

The transcriptomics data revealed that only eight, three and one gene were uniquely expressed in the control, Lac and LacHyd groups, respectively (Figure 1B). Comparing the Lac group to the control group revealed 79 upregulated and 83 downregulated genes. Comparing the LacHyd group to the control group revealed 65 upregulated and 68 downregulated genes. This was unsurprising given that *G. vaginalis* did not have a complex signaling system. Surprisingly, the comparison of the Lac group to the LacHyd group revealed only one upregulated and six downregulated genes. The similar stress response under the two conditions suggests that the lactate stimulation may represent the lactobacilli stimulation in *G. vaginalis* (Appendix A). The genes with significant differences are listed in Table 1 and Appendix A. The RT-PCR was used to confirm the expression levels of the eight virulence genes, revealing that, in addition to the stress response genes being upregulated, the genes related to the epithelial cell adhesion and biofilm formation were actually downregulated (Figure 2). This could be the mechanism by which the lactobacilli probiotics inhibit GV cell adhesion to the epithelial cells via biofilm.

### 3.3. The Proteomics Data Are Consistent with the Transcriptomic Data

The TMT quantitative proteomics were performed on the LacHyd group to determine whether the protein regulation was consistent with the transcription level after stimulation. The results revealed that 2740 proteins were identified at the species level, which was more than twice the number of the genome genes, indicating the species’ high genetic heterogeneity. According to the numbers of the differentially expressed proteins, the standard screening of the proteins with differential expression multiples was adjusted to more than 1.2 times (upregulated by more than 1.2 times or downregulated by less than 0.83 times) and a *p* value of <0.05. There were 86 upregulated differentially expressed proteins and 55 downregulated differential expression proteins in the LacHyd group compared to the control group. The analysis of the GO and KEGG pathways revealed that the function of these differentially expressed proteins was primarily involved in the processes of metabolism, cell division, localization, response to stimulus and transcriptional regulation. Since the main pathogenic factors found in the upregulated proteins were still efflux transporters, it was hypothesized that ATCC 14,019 actively effluxed harmful ions and small molecules by upregulating transporters (HMPREF0421_20219 and HMPREF0421_21025). Furthermore, while some virulence factors were upregulated, the biofilm formation-related proteins were downregulated (Table 2), implying that GV was intended to gather weapons and energy to combat the lactobacilli whether or not it worked.

## 4. Discussion

Many upregulated genes of the strain ATCC 14,019 in the LacHyd and Lac groups belonged to bacterial efflux pumps, including the ABC transporter, MFS transporter, sulfonate ABC transporter osmotic enzyme and cobalt ABC transporter (HMPREF0421_21024, HMPREF0421_20219, HMPREF0421_21221 and HMPREF0421_20844). The ABC transporters used ATP hydrolysis to provide energy for the cells, and their upregulation helped in the removal of harmful substances via the ABC export system (HMPREF0421_21024) [20,21]. The MFS transporter was most commonly described as an efflux pump that removes antibiotics (e.g., TetB and TetK) from the cells [22,23,24]. Various families were responsible for the efflux of various types of small molecules. This gene was only upregulated in the LacHyd group, which was thought to be responsible for the hydrogen peroxide efflux (HMPREF0421_20219). A small molecular compound and intracellular ion efflux was mediated by the cobalt transporters (HMPREF0421_20844) [25]. It is worth noting that histidine kinase, as one of the strain’s six pairs of two-component systems, was found to be highly upregulated in the Lac and LacHyd groups, implying that it could be responsible for the lactate signal transduction on the cell membrane (HMPREF0421_21022). Furthermore, 3-Oxoacetyl-ACP reductase was involved in the synthesis of siderophores, and its upregulation indicated that the strain can obtain more iron from the environment to inhibit the lactobacilli iron uptake (HMPREF0421_21015) [15,26]. The ATCC 14,019 downregulated genes were mostly involved in the adhesion and biofilm syntheses (HMPRE0421_21089, HMPRE0421_20208, HMPRE0421_20303 and HMPRE0421_20991; Figure 2). *G. vaginalis* can promote cell aggregation via the biofilm formation and adhere to the host cells via pili and surface-anchored proteins. The gene expression level of the Lac group was nearly identical to that of the LacHyd group, with only five different genes, and the fold change was less than 2 (Appendix A). In other words, hydrogen peroxide had little antimicrobial effect in mixing with lactate, which was consistent with the conclusion of a previous study [27], implying that the strains took response measures to the lactobacilli signals and dealt with more risks based on lactate.

Apart from the efflux transporters, the expressional level of vaginolysin, a cholesterol-dependent hemolysin (HMPREF0421_20066), had been upregulated by 1.43 times according to the proteomics data [28]. CHAP is a protein domain found in many extracellular proteins and receptors. It was initially identified in enzymes that degraded bacterial cell walls, and it recognizes the polysaccharides involved in N-acetylglucosamine synthesis, including peptidoglycan (HMPREF0421_20542) [29]. When stimulated, the virulence factors increased their competitive capacity against the lactobacilli. The pathogenic factors found in the downregulated proteins included the pili, biofilm syntheses and adhesion proteins, which were consistent with the transcriptomics results. The PTS EII transmembrane protein, which was mostly downregulated, was primarily responsible for the glucose transport and was critical for maintaining normal cell growth and biofilm synthesis (HMPREF0421_20893) [30]. RelE, a toxin–antitoxin pair encoded by the RelBE operon, was a very effective translation inhibitor in vivo and in vitro. RelB can reverse the RelE inhibitory effects on the protein synthesis in vivo, and RelE plays a regulatory role on bacteria in the process of adapting to poor growth conditions and forming persistent cells by controlling the cell dormancy (HMPREF0421_21052) [31]. The maltose-binding protein was responsible for the maltodextrin absorption (HMPREF0421_20232), which provided raw materials for the biofilm synthesis. The sugar ABC transporters transported monosaccharides or oligosaccharides, providing oligosaccharides units for cell growth or biofilm formation (HMPREF0421_20098) [32]. Therefore, we can speculate that, while the formed biofilm increased the *G. vaginalis* tolerance to lactate and hydrogen peroxide [33], the lactobacilli were able to inhibit *G. vaginalis* cell growth and biofilm formation in vivo.

In conclusion, in a healthy vaginal microecological environment, the lactobacilli, as the dominant flora, secrete a high concentration of lactate and hydrogen peroxide to the extracellular environment, forcing *G. vaginalis* to expend considerable energy to excrete these two small molecules. There is no excess energy for its own growth and proliferation (Figure 3A). When the vaginal microecology is out of balance, the number of lactobacilli and the concentrations of lactate and hydrogen peroxide are significantly reduced. Anaerobic microorganisms, primarily *G. vaginalis*, begin to upregulate the expression levels of the pili, adhesion proteins and biofilm formation, and gradually form biofilm adhered to the vaginal epithelial cells, resulting in BV (Figure 3B). Therefore, inhibiting the expression of the adhesion proteins and biofilm components using *G. vaginalis* at the onset of a vaginal microecological imbalance could be the key to preventing bacterial vaginitis.

## Figures and Tables

**Figure 1 microorganisms-11-00695-f001:**
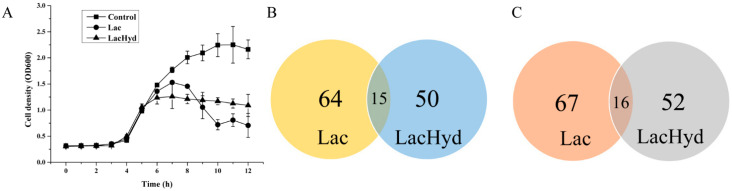
General comparison of the control, Lac group and LacHyd group. (**A**) Growth curves of the three groups. (**B**) Venn analysis of the shared genes that were upregulated compared to the control in the two groups. (**C**) Venn analysis of the shared genes that were downregulated compared to the control in the two groups. The pH values of the three groups were 7.0., 5.5 and 5.5, respectively; the H_2_O_2_ concentration of the LacHyd group was 0.5 mM.

**Figure 2 microorganisms-11-00695-f002:**
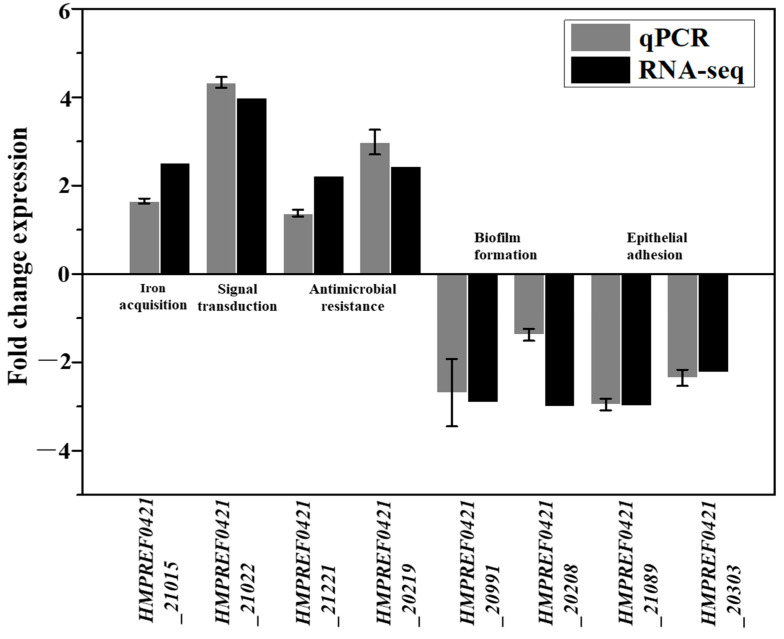
Quantification of the transcription of the virulence genes that were expressed differentially in *G. vaginalis* cultured under lactate and hydrogen peroxide. Bars represent the mean, and the error bars the standard error of the mean (mean ± SEM).

**Figure 3 microorganisms-11-00695-f003:**
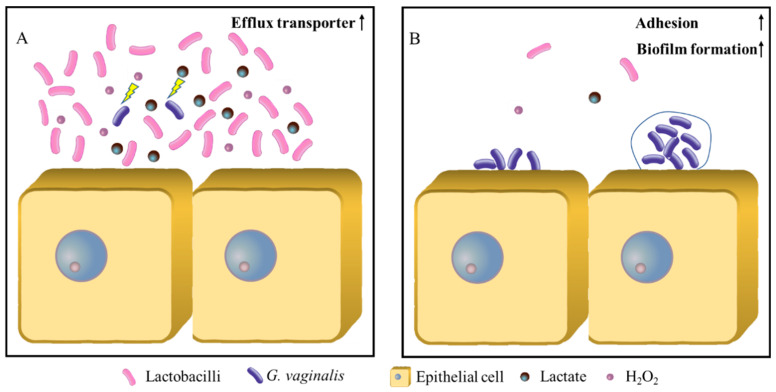
The effect model of the lactobacilli on the biofilm formation of *G. vaginalis*. (**A**) Healthy vaginal microecosystem; (**B**) BV vaginal microecosystem.

**Table 1 microorganisms-11-00695-t001:** List of the 10 genes with the highest fold change values among the differentially expressed genes in *G. vaginalis* cultured under lactate and hydrogen peroxide. All *p* values were less than 0.02.

GeneID	Log2Fold Change	*p* Value	*P* adj	Function
Upregulated				
HMPREF0421_21022	3.99	2.91 × 10^−206^	3.00 × 10^−204^	Sensor histidine kinase
HMPREF0421_21024	3.90	8.07 × 10^−31^	5.63 × 10^−30^	ABC transporter
HMPREF0421_21015	2.52	6.07 × 10^−62^	1.04 × 10^−60^	3-oxoacyl-ACP reductase
HMPREF0421_20839	2.47	0	0	NADH dehydrogenase
HMPREF0421_20219	2.44	1.04 × 10^−42^	1.11 × 10^−41^	Major facilitator superfamily (MFS) transporter
HMPREF0421_21221	2.22	0	0	Sulfonate ABC transporter permease
HMPREF0421_20844	2.19	7.77 × 10^−171^	6.12 × 10^−169^	Cobalt ABC transporter
HMPREF0421_20940	2.18	0	0	Oligoribonuclease
HMPREF0421_21136	2.07	3.65 × 10^−248^	6.12 × 10^−246^	Nicotinate phosphoribosyltransferase
HMPREF0421_20795	2.03	6.02 × 10^−40^	5.64 × 10^−39^	Phosphoserine phosphatase SerB
Downregulated				
HMPREF0421_20208	−3.01	0	0	Hydroxyethylthiazole kinase
HMPREF0421_21089	−2.99	7.76 × 10^−15^	2.90 × 10^−14^	Type IV prepilin peptidase
HMPREF0421_20991	−2.91	5.60 × 10^−25^	3.11 × 10^−24^	Transferring glycosyl groups
HMPREF0421_21002	−2.90	4.39 × 10^−79^	1.09 × 10^−77^	Cell division protein FtsK
HMPREF0421_21216	−2.45	1.91 × 10^−119^	9.13 × 10^−118^	Ribonuclease HII
HMPREF0421_20331	−2.40	4.83 × 10^−258^	7.35 × 10^−57^	Serine/threonine protein kinase
HMPREF0421_20297	−2.33	8.19 × 10^−91^	2.49 × 10^−89^	Shikimate kinase
HMPREF0421_20303	−2.23	5.91 × 10^−212^	6.60 × 10^−210^	Actinobacterial surface-anchored domain protein
HMPREF0421_20209	−2.11	5.75 × 10^−224^	8.56 × 10^−222^	Thiamine biosynthetic process
Novel C3	−2.05	8.18 × 10^−16^	3.14 × 10^−15^	Transcription regulator

**Table 2 microorganisms-11-00695-t002:** List of the 10 genes with the highest fold change values among the differentially expressed proteins in *G. vaginalis* cultured under lactate and hydrogen peroxide. All *p* values were less than 0.05.

GeneID	Log2Fold Change	*p* Value	Function
Upregulated			
HMPREF0421_20219	2.69	0.000234	Major facilitator superfamily (MFS) transporter
HMPREF0421_20371	2.39	0.001399	CDP-diacylglycerol-glycerol-3-Phosphate 3-phosphatidyltransferase
HMPREF0421_20787	1.50	0.008551	Prevent-host-death family antitoxin
HMPREF0421_20066	1.43	0.020726	Vaginolysin
HMPREF0421_20542	1.43	0.000758	CHAP domain protein (lytic ability)
HMPREF0421_20467	1.42	0.006055	Cell division protein FtsI
HMPREF0421_21025	1.32	0.002046	ABC transporter ATP-binding protein
HMPREF0421_20164	1.24	0.005587	Heat shock protein HtpX
HMPREF0421_20745	1.22	0.002437	Methylenetetrahydrofolate reductase
HMPREF0421_21364	1.21	0.004129	Response regulator
Downregulated			
HMPREF0421_20893	0.65	0.000724	PTS system transporter subunit IIC
HMPREF0421_20297	0.73	0.032017	Shikimate kinase
HMPREF0421_20633	0.74	0.000496	Formate acetyltransferase
HMPREF0421_20235	0.76	0.009039	Sugar ABC transporter membrane protein
HMPREF0421_20979	0.76	0.001612	TM2 domain-containing protein
HMPREF0421_20098	0.78	0.001898	Sugar ABC transporter membrane protein
HMPREF0421_20232	0.81	0.003311	Maltose-binding protein
HMPREF0421_20422	0.81	0.025688	Exopolyphosphatase
HMPREF0421_20575	0.82	0.000581	Phosphate acetyltransferase
HMPREF0421_21052	0.83	0.022516	RelE-family TA system toxin

## Data Availability

The transcriptomics data have been submitted to NCBI with the Accession No. PRJNA865867.

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
