# Peer review of "Transcriptomic and Proteomic Analysis of *Gardnerella vaginalis* Responding to Acidic pH and Hydrogen Peroxide Stress"

_microorganisms, 2023, doi:10.3390/microorganisms11030695_

Round 1
Reviewer 1 Report (Previous Reviewer 3)
I am satisfied with the revisions made
Author Response
Thank you for your review.
Reviewer 2 Report (Previous Reviewer 2)
The manuscript: Transcriptomic and proteomic analysis of Gardnerella vaginalis responding to acidic pH and hydrogen peroxide stress
Kundi Zhang , Mengyao Lu, Yuxin Qiu, Xiaoxuan Zhu, Hongwei Wang, Yan Huang, Hongjie Dong,Lichuan Gu
Title: corresponds to the content of the article
Introduction: enough; represent the essence of the problem; does not require change
Studies to determine the pathogenesis of bacterial vaginosis in connection with the disappearance of lactobacilli producing hydrogen peroxide and lactic acid, and to establish the leading role of Gardnerella vaginalis are of great importance. Therefore, this study is of great practical and scientific interest. Moreover, despite the treatment with metronidazole or clindamycin, there are frequent recurrences of BV. The chosen research direction is extremely relevant. The authors selected shk (sensor histidine kinase) encoding a membrane protein related to the regulation of biofilm formation, and trx (thioredoxin-disulfide reductase) encoding an intracellular protein as markers.
Methodology: meets the requirements of the journal and the branch of knowledge; does not require modification;
Statistics: meets the requirements of international standards
Results: formulated clearly and clearly. Supplemented with tables and figures and reinforced with statistical data.
Discussion: sufficient and consistent with the main results obtained.
Literature: sufficient for the article and does not require additions Article design: meets the requirements of the journal
Conclusion: the article meets the requirements of the journal and can be published without significant revision.

Author Response
Thank you for your review.
Reviewer 3 Report (New Reviewer)
Zhang and coworkers describe the impact of lactate and hydrogen peroxide on the gene and protein expressions of Gardnerella vaginalis. The authors showed that the lactate and lactate+ H2O2 stress had a limited impact on the bacterial gene and protein expression.
There are questions that should be addressed.
1 1. A major problem is that the authors did not mention the statistical analyses for the gene expression and proteomics data. In order to be accepted the missing statistics should be included, including FDR calculations.
2 2. Also, the fold change threshold is not clear for the gene expression data, only the protein expression data. For the proteomics data the authors chose a rather low (1.2 fold) up/downregulation fold change threshold which is rather small. Why did the authors chose this threshold level? This should be included in the results/ discussion part.
3 3. It is hard to understand (row 191-193). “The results revealed that 2,740 proteins were identified at the species level, which is more than twice the number of genome genes, indicating the species’ high genetic heterogeneity.”
4 4. Figure 1B Venn diagram is misleading, because it shows the unique genes in the samples. A more common Venn diagram of the up and down-regulated genes would be much better here.
Altogether after the corrections, the manuscript may be acceptable for publication.
Author Response
Zhang and coworkers describe the impact of lactate and hydrogen peroxide on the gene and protein expressions of Gardnerella vaginalis. The authors showed that the lactate and lactate+ H2O2 stress had a limited impact on the bacterial gene and protein expression.
There are questions that should be addressed.
- A major problem is that the authors did not mention the statistical analyses for the gene expression and proteomics data. In order to be accepted the missing statistics should be included, including FDR calculations.
>We have added the FDR values of transcriptomic data showed as “P adj” in Table 1, S3&S4. In addition, as the settings of proteomics assay were relatively strict and the TMT-marked methods had correction effect itself, the FDR values were not provided.
- Also, the fold change threshold is not clear for the gene expression data, only the protein expression data. For the proteomics data the authors chose a rather low (1.2 fold) up/downregulation fold change threshold which is rather small. Why did the authors chose this threshold level? This should be included in the results/ discussion part.
>In general, the difference of protein expression level between control and LacHyd group was not apparent very much, so we chose “1.2 fold” to get more differentially expressed proteins. We have explained it in Lines 198-199.
- It is hard to understand (row 191-193). “The results revealed that 2,740 proteins were identified at the species level, which is more than twice the number of genome genes, indicating the species’ high genetic heterogeneity.”
>There were many hypothetical proteins in the annotated genome of G. vaginalis ATCC 14019, so we used the whole G. vaginalis (GV) genome database including genome sequences of all GV strains to identify the proteins. As the GV strains were of high genetic heterogeneity, it’s possible that one peptide fragment can be identified as various proteins.
- Figure 1B Venn diagram is misleading, because it shows the unique genes in the samples. A more common Venn diagram of the up and down-regulated genes would be much better here.
>We have painted new Venn diagrams showing up and down-regulated genes to replace the old one as Figure 1B&1C.
This manuscript is a resubmission of an earlier submission. The following is a list of the peer review reports and author responses from that submission.
Round 1
Reviewer 1 Report
The authors cultivated a single G.vaginalis strain (ATCC 14019), then the culture was alkalized to pH 5.5 or supplemented with 0.5mM lactic acid. Using transcriptomic and proteomic techniques the authors analyzed the up/down-regulation of the genes
It is a poorly written paper: the authors are not properly familiar with the object of the study, i.e. vaginal microbiota, particularly Gardnerella spp. Comments on the Introduction section are below. The English language needs extensive editing. In the Abstract (lines 14-16), the authors indicated Gardnerella co-culturing with lactobacilli, however, this does not find a response in the body text. The authors did not justify the selected concentration of 0.5mM lactate added to G.vaginalis culture. Moreover, they did not critically discuss the impact of hydrogen peroxide on the „health“ of vaginal microbiota argued in the paper by Ravel and colleagues. (https://microbiomejournal.biomedcentral.com/articles/10.1186/s40168-018-0418-3). The last paragraph of the discussion section (lines 247-258) is unacceptable in all aspects. The transcriptomic and proteomic analysis is fine, however, what is the reason to analyze gene expression under scientifically unjustified conditions?
Introduction:
1 1. Lines 30-33: the authors lead to a misleading message: BV is one of the forms of the imbalance of vaginal microbiota, but not the single one. The paragraph should be revised.
2 2. Line 31: please rename „flora“ to „microbiota.“
33. Lines 34-35: complicated with complications...
44. Lines 37-41: it is complicated to understand the meaning of the sentences. No references prove the statements.
55. Line 47, Abstract: Gardnerella vaginalis is not a pathogen. It is a commensal of vaginal microbiota.
66. G.vaginalis is only one species of Gardnerella most probably linked with the pathogensis of BV. Why the authors selected this species for the study?
Methods:
1. Lines 65-67: please, indicate the company, H2O2 and lactate were obtained from. What was OD600 or CFU of G.vaginalis culture when lactate/ H2O2 were added?
2. Why these markers shk (sensor histidine kinase) and trx (thioredoxin-disulfide reductase) were selected? How these markers respond to lactic acid & H2O2. References?
3. Line 65: what does mean „cultivated earth exponential phase”?
Results:
1 Why H2O2 concentration of 0.5mM was selected? What is the concentration of H2O2 produced by lactobacilli in vivo? What is the concentration of H2O2 produced by lactobacilli strains in vitro? See ref. https://pubmed.ncbi.nlm.nih.gov/16965352/,
Author Response
The authors cultivated a single G.vaginalis strain (ATCC 14019), then the culture was alkalized to pH 5.5 or supplemented with 0.5mM lactic acid. Using transcriptomic and proteomic techniques the authors analyzed the up/down-regulation of the genes
It is a poorly written paper: the authors are not properly familiar with the object of the study, i.e. vaginal microbiota, particularly Gardnerella spp. Comments on the Introduction section are below. The English language needs extensive editing. In the Abstract (lines 14-16), the authors indicated Gardnerella co-culturing with lactobacilli, however, this does not find a response in the body text. The authors did not justify the selectd concentration of 0.5mM lactate added to G.vaginalis culture. Moreover, they did not critically discuss the impact of hydrogen peroxide on the „health“ of vaginal microbiota argued in the paper by Ravel and colleagues. (https://microbiomejournal.biomedcentral.com/articles/10.1186/s40168-018-0418-3). The last paragraph of the discussion section (lines 247-258) is unacceptable in all aspects. The transcriptomic and proteomic analysis is fine, however, what is the reason to analyze gene expression under scientifically unjustified conditions?
>Thank you for your suggestions. In the Abstract (lines 20-22), we indicated that “In this study, lactate and hydrogen peroxide were added to G. vaginalis culture medium to simulate the co-culture of lactobacilli and G. vaginalis”, which means the experiment is just a simulation of co-culturing Gardnerella and lactobacilli. According to the clinic standards of vaginal microecological balance (The human vaginal microbial community - PubMed (nih.gov)), the pH value range of the healthy vagina should be 3.5-4.5, but Gardnerella didn‘t grow when the pH value was below 5.0, so we adjusted the pH value to 5.5 with lactate in order to get the sub-lethal Gardnerella cells. For the same reason, hydrogen peroxide was added to a final concentration of 0.5 mM to get the sub-lethal state according to that the production of lacotobacilli was 0.05-1.0 mM in vitro (The human vaginal microbial community - PubMed (nih.gov)). We noticed that low concentration hydrogen peroxide didn‘t inhibit the cell growth at all, indicating that hydrogen peroxide actually has little antimicrobial effect, which was consistent with the conclusion of the reference Raval et. al. We have added some ststaments in Lines 239-242. In addition, we also improved English writting of the paper by an English-native speaker.
Introduction:
1 1. Lines 30-33: the authors lead to a misleading message: BV is one of the forms of the imbalance of vaginal microbiota, but not the single one. The paragraph should be revised.
> We have modified the description as “A loss of vaginal lactobacilli and an overgrowth of diverse anaerobes are two characteristics of vaginal microecological imbalance” in Lines 36-37.
2 2. Line 31: please rename „flora“ to „microbiota.“
> Corrected in Line 38.
3 3. Lines 34-35: complicated with complications...
> We revise the sentence as “ Patients can develop complications whether or not they have clinical symptoms” in Lines 40-41.
4 4. Lines 37-41: it is complicated to understand the meaning of the sentences. No references prove the statements.
> The reference is added in Line 45.
5 5. Line 47, Abstract: Gardnerella vaginalis is not a pathogen. It is a commensal of vaginal microbiota.
>G. vaginalis is a species with high genetic heterogeneity in strains from different patients, which had been verified in Yeoman et.al (https://www.ncbi.nlm.nih.gov/pubmed/20865041) and our previous study Antibiotic resistance and pathogenicity assessment of various Gardnerella sp. strains in local China - PubMed (nih.gov). The genomes and characters of some GV strains in healthy host always differ widely from the strains isolated from patients, with only 70% similarity in genome sequence. They are quite different in antibiotic resistance, biofilm formation abilities. For most BV patients, GV is still the main pathogen as it forms biofilm which attaches to the epithelial cells.
6 6. G.vaginalis is only one species of Gardnerella most probably linked with the pathogensis of BV. Why the authors selected this species for the study?
>Our lab has been working on the mechanism of biofilm for many years. In these years many gynaecologists complain that it is a very big challenge for them to treat GV infection because of its biofilm. In order to solve this problem, we started to study on the biofilm formation, antibiotic resistance, virulence factors of GV and so on.
Methods:
- Lines 65-67: please, indicate the company, H2O2 and lactate were obtained from. What was OD600 or CFU of G.vaginalis culture when lactate/ H2O2 were added?
>We have added the supplier of lactate and H2O2 in Lines 73-74 and “when cell density reached an OD600 of approximately 1.0” in Lines 72-73.
- Why these markers shk (sensor histidine kinase) and trx (thioredoxin-disulfide reductase) were selected? How these markers respond to lactic acid & H2O2. References?
>Shk is located in the cell membrane and is belong to the two-component system (TCS), which is related to the regulation of biofilm formation(Frontiers | Two-Component Signal Transduction Systems: A Major Strategy for Connecting Input Stimuli to Biofilm Formation (frontiersin.org)). Biofilm formation was regulated by c-di-A/GMP, quorum sensing and TCS. However, GV lacked the former two system, so we focused on its TCS. Ldh (lactate dehydrogenase) and Trx are intracellular proteins which response to lactic acid and H2O2 (Molecular mechanism involved in the response to hydrogen peroxide stress in Acinetobacter oleivorans DR1 - PubMed (nih.gov)), respectively, but the expressional level of ldh remained stable even after adding lactic acid for 1h(data not shown), so we chose shk and trx as markers to indicate the response of GV to the stress. I have added references in Line 78.
- Line 65: what does mean „cultivated earth exponential phase”?
> Corrected as “early exponential phase” in Line 72.
Results:
1 Why H2O2 concentration of 0.5mM was selected? What is the concentration of H2O2 produced by lactobacilli in vivo? What is the concentration of H2O2 produced by lactobacilli strains in vitro? See ref. https://pubmed.ncbi.nlm.nih.gov/16965352/,
> The range of H2O2 concentration (0.05-1.0 mM) was referred from The human vaginal microbial community - PubMed (nih.gov), which was the same as the reference you supplied. 0.5 mM was selected to get the sub-lethal GV cells, and we have added the references in Line 75. Actually, we haven't looked up any reference described the the concentration of H2O2 produced by lactobacilli in vivo, maybe due to the difficulty of measuring.
Reviewer 2 Report
The manuscript: Transcriptomic and proteomic analysis of Gardnerella vaginalis responding to acidic pH and hydrogen peroxide stress
Kundi Zhang , Mengyao Lu, Yuxin Qiu, Xiaoxuan Zhu, Hongwei Wang, Yan Huang, Hongjie Dong,Lichuan Gu
Title: corresponds to the content of the article
Introduction: enough; represent the essence of the problem; does not require change
Methodology: meets the requirements of the journal and the branch of knowledge; does not require modification;
Statistics: meets the requirements of international standards
Results: clearly stated and do not require changes
Discussion of the results: sufficient and consistent with the main results obtained.
Literature: sufficient for the article and does not require additions
Article design: meets the requirements of the journal
Conclusion: the article meets the requirements of the journal and can be published without significant revision
Author Response
The manuscript: Transcriptomic and proteomic analysis of Gardnerella vaginalis responding to acidic pH and hydrogen peroxide stress
Kundi Zhang , Mengyao Lu, Yuxin Qiu, Xiaoxuan Zhu, Hongwei Wang, Yan Huang, Hongjie Dong,Lichuan Gu
Title: corresponds to the content of the article
Introduction: enough; represent the essence of the problem; does not require change
Methodology: meets the requirements of the journal and the branch of knowledge; does not require modification;
Statistics: meets the requirements of international standards
Results: clearly stated and do not require changes
Discussion of the results: sufficient and consistent with the main results obtained.
Literature: sufficient for the article and does not require additions
Article design: meets the requirements of the journal
Conclusion: the article meets the requirements of the journal and can be published without significant revision
>Thank you for your review. In addition, we improved English writting of the paper by an English-native speaker.
Reviewer 3 Report
This is a straightforward study simulating co-culture of Gardnerella vaginalis, a key bacterial vaginosis (BV) bacterium, and lactobacilli via culturing G. vaginalis in the presence of lactate and hydrogen peroxide, to identify G. vaginalis genes responding to acidic pH and hydrogen peroxide stress. These experiments showed that the majority of up-regulated genes were related to the efflux, while down-regulated genes were mostly those involved in biofilm formation and epithelial cell adhesion, which may have implication for the development of new BV drugs.
The study is methodologically sound, clearly presented. I only have some minor suggestions:
1) In the Introduction, I would suggest omitting the data on the ethnical differences in the vaginal microbiome as not so relevant, and instead focusing a bit more on BV treatment problems and options.
2) Line 38: the prevalence figures need referring to the source.
3) Line 49: “normal vagina” should be changed for “healthy vagina”.
4) In the Discussion, I would suggest mentioning the study showing that G. vaginalis biofilms are manifold more tolerable to high concentrations of lactate and hydrogen peroxide than planktonic cultures (Patterson JL, Girerd PH, Karjane NW, Jefferson KK. Effect of biofilm phenotype on resistance of Gardnerella vaginalis to hydrogen peroxide and lactic acid. Am J Obstet Gynecol. 2007 Aug;197(2):170.e1-7. doi: 10.1016/j.ajog.2007.02.027. PMID: 17689638; PMCID: PMC2020809).
Author Response
This is a straightforward study simulating co-culture of Gardnerella vaginalis, a key bacterial vaginosis (BV) bacterium, and lactobacilli via culturing G. vaginalis in the presence of lactate and hydrogen peroxide, to identify G. vaginalis genes responding to acidic pH and hydrogen peroxide stress. These experiments showed that the majority of up-regulated genes were related to the efflux, while down-regulated genes were mostly those involved in biofilm formation and epithelial cell adhesion, which may have implication for the development of new BV drugs.
>Thank you for your review. In addition, we improved English writting of the paper by an English-native speaker.
The study is methodologically sound, clearly presented. I only have some minor suggestions:
- In the Introduction, I would suggest omitting the data on the ethnical differences in the vaginal microbiome as not so relevant, and instead focusing a bit more on BV treatment problems and options.
> We have deleted the statement of the ethnical differences and added some BV treatment data in Lines 47-51.
2) Line 38: the prevalence figures need referring to the source.
> The reference has been included in Line 45.
3) Line 49: “normal vagina” should be changed for “healthy vagina”.
> Corrected in Line 55.
4) In the Discussion, I would suggest mentioning the study showing that G. vaginalis biofilms are manifold more tolerable to high concentrations of lactate and hydrogen peroxide than planktonic cultures (Patterson JL, Girerd PH, Karjane NW, Jefferson KK. Effect of biofilm phenotype on resistance of Gardnerella vaginalis to hydrogen peroxide and lactic acid. Am J Obstet Gynecol. 2007 Aug;197(2):170.e1-7. doi: 10.1016/j.ajog.2007.02.027. PMID: 17689638; PMCID: PMC2020809).
> The relevant content has been added as you suggested in Lines 263-265.
Round 2
Reviewer 1 Report
NA